# Exposure to pollutants for household cooking and lighting and pediatric post-discharge mortality following a severe infection in Uganda

Gurvir S. Dhutt[1,2], Cherri Zhang[1], Elias Kumbakumba[3], Abner Tagoola[4], Peter Moschovis[5], Stephen Businge[6], Niranjan Kissoon[1,2], Nathan Kenya Mugisha[7], Jerome Kabakyenga[8,9], Matthew O. Wiens[1,2]*

1 Institute for Global Health, BC Children's Hospital Research Institute, Vancouver, British Columbia, Canada, 2 Faculty of Medicine, University of British Columbia, Vancouver, British Columbia, Canada, 3 Department of Paediatrics and Child Health, Mbarara University of Science and Technology, Mbarara, Uganda, 4 Department of Paediatrics, Jinja Regional Referral Hospital, Jinja City, Uganda, 5 Division of Global Health, Massachusetts General Research Institute, Boston, Massachusetts, United States of America, 6 Holy Innocents Children's Hospital, Mbarara, Uganda, 7 Walimu, Kampala, Uganda, 8 Maternal Newborn and Child Health Institute, Mbarara University of Science and Technology, Mbarara, Uganda, 9 Department of Community Health, Faculty of Medicine, Mbarara University of Science and Technology, Mbarara, Uganda

* Matthew.Wiens@bcchr.ca

## Abstract

### Background

Particulate matter from household air pollution (HAP) is linked to half of all lower respiratory infection deaths among children under 5 years of age. In rural Uganda, similar number of children die 6-months post discharge as during hospitalization for severe infections. However, it is unclear whether exposure to HAP contributes to poor health and death after discharge. We investigated the association between cooking and household lighting practices and mortality 6-months post-discharge among children under 5 years of age treated for severe infection in rural Uganda.

### Methods

We conducted a secondary analysis of data from observational cohort studies, conducted between July 2017 to July 2021, among 6,955 children 0 to 5 years admitted to one of six Ugandan hospitals for a severe infectious illness. Clinical signs and symptoms, sociodemographic variables, and mortality up to 6-months post-discharge was collected for all participants, and follow-up rates were >95%. Exposure variables included type of cooking fuel used, location of cooking (e.g. indoors, outdoors), and primary source of household lighting. We assessed post-discharge mortality using simple and multivariate Poisson regression.

**Data availability statement:** Regarding data sharing, due to the sensitive nature of the patient data and the risk for re-identification, data is only available through reasonable request to the Pediatric Sepsis Data CoLaboratory. All data and collection tools are available through the Smart Discharges Dataverse. The dataset is published and can be viewed using this link: https://doi.org/10.5683/SP3/ZLZQOG (Citation: Dhutt GS, Zhang C, Kumbakumba E, Tagoola A, Moschovis P, Businge S, et al. Exposure to pollutants for household cooking and lighting and pediatric post-discharge mortality following a severe infection in Uganda [Internet]. V1 ed. Borealis; 2025. Available from: https://doi.org/10.5683/SP3/ZLZQOG ). Access to the data can be requested through the link above. Access to data is granted on a case-by-case basis following approval from the authors and the Data Governance Committees. The CoLab coordinator can be reached at sepsiscolab@bcchr.ca. Alternative contact is Mia Sheehan who can be reached at mia.sheehan@bcchr.ca.

**Funding:** This work was supported by Health Research BC, award number SCH-2021-1581, to MW. The funders had no role in study design, data collection and analysis, decision to publish, or preparation of the manuscript.

**Competing interests:** The authors have declared that no competing interests exist.

## Results

The unadjusted risk ratio of 6-month post-discharge mortality by dual or single exposure to pollutant fuel sources for cooking indoors and household lighting, when compared to minimal exposure, was 1.57 (95%CI 1.17, 2.11) and 1.20 (95%CI 0.94, 1.54), respectively. Adjusting for age, sex, distance to hospital, maternal education, and maternal HIV status, the adjusted risk ratios for dual and single exposure became 1.30 (95%CI 0.96, 1.76) and 1.08 (95%CI 0.84, 1.38). There was no significant interaction between exposure and age, sex, maternal education, or anemia status.

## Discussion

This analysis did not find a statistically significant association between HAP and 6-month post-discharge mortality. However, HAP cannot be ruled out as a contributor in this population where malnutrition, comorbidities and sociodemographic vulnerabilities are common.

## Introduction

According to the World Health Organization (WHO), about one-third (2.3 billion) of the global population, majority of whom live in low- and middle-income countries (LMIC), use kerosene and coal as fuel for open fires or inefficient stoves [1]. This practice results in harmful indoor air pollution and, in 2020, was reported to be associated with about 3.2 million deaths, disproportionately impacting children in Sub-Saharan Africa due to increased pollution levels resulting from rapid urbanization and industrialization in this region [1–4].

Household air pollution leads to impaired child lung development and recurrent childhood respiratory infections [5]. Children are especially vulnerable given their immature cardiorespiratory system and higher respiratory and heart rates which aid the deposition of particulate matter into their tissues and bloodstream [6,7]. Children further face increased risk due to extended exposure to particulate matter from household air pollution from engagement in household activities such as cooking and household lighting [1,6]. In the Sub-Saharan African context, the use of household pollutant fuel sources (e.g., kerosene or coal) has been shown to increase the risk of all-cause child mortality, with current estimates being as high as a 126% increase in mortality due to pollutant cooking fuel source use inside the household [8–14].

In many LMIC within the Sub-Saharan Africa and Southern Asia regions, deaths following hospital discharge accounts for about half of all deaths for acute illnesses in children under 5 [15–17]. While the long-term impact of household air pollution has been the focus of significant research, it remains unclear how exposure to pollutant sources of cooking and household lighting fuel during the vulnerable post discharge period impacts a child's ability to recover over a shorter term period.

Therefore, the purpose of this study was to understand the relationship between cooking and lighting behaviours and post-discharge outcomes. For this reason, we investigated the association between exposure to pollutant fuel sources for cooking and lighting and post-discharge mortality among children under 5 years of age treated for severe infection in rural Uganda.

## Methods

### Study designs and approval

This study conducted a secondary analysis of data from a prospective observational cohort study among children admitted to hospitals in Uganda for suspected sepsis who were aged 0 to 60 months at the time of admission [16]. Briefly, participants with suspected sepsis were recruited from 6 hospitals (Mbarara Regional Referral Hospital, Mbarara, Southwest Uganda; Holy Innocents Children's Hospital, Mbarara, Southwest Uganda; Masaka Regional Referral Hospital, Masaka, Central Uganda; Jinja Regional Referral Hospital, Jinja City, East Uganda; Villa Maria Hospital, Masaka, Central Uganda; and Uganda Martyrs Hospital, Ibanda, Southwest Uganda) between July 13, 2017, to July 28, 2021. These facilities serve approximately 8.2 million individuals over 30 districts, including approximately 1.4 million children who were aged 5 years or younger during the study period [18]. Participants were followed from the time of admission to 6-months post-discharge. Follow-up rates were >95% for all cohorts.

These studies were approved by the Mbarara University of Science and Technology Research Ethics Committee (15/10–16, 27-Jan-2017), and the University of British Columbia–Children and Women's Health Centre of British Columbia Research Ethics Board (H16–02679, 09-May-2017). This manuscript adheres to the guidelines for Strengthening the Reporting of Observational Studies in Epidemiology (STROBE) [19].

### Study population inclusion and exclusion criteria

The present analysis included participants who were admitted to the hospital with suspected sepsis, which was defined as having a confirmed or suspected infection determined by the treating medical team. About 90% of children admitted to a Ugandan hospital for presumptive or confirmed infection meet the International Pediatric Sepsis Consensus Conference (IPSCC) definition for sepsis [20]. The IPSCC defines sepsis as a confirmed or suspected infection with the presence of a systemic inflammatory response syndrome.

Children admitted for a short-term observation period (<24 hours), trauma, previously enrolled, those who resided outside of the hospital catchment area, or newborns who had not been discharged home were excluded. Parents or legal guardians of all study participants were asked to provide written informed consent.

### Data collection

Data collection procedures have been described in detail previously [16]. Trained study nurses collected clinical, sociodemographic, and outcome data from all consented participants. Clinical data included vital signs, clinical signs and symptoms and comorbidities. Laboratory parameters included point of care hemoglobin, malaria parasitemia, serum lactate, human immunodeficiency virus (HIV) status and serum glucose. Sociodemographic data included maternal age, education, HIV status, distance of home from facility, type of cooking fuel used, location of cooking and primary source of household lighting. Each participant's discharge status was obtained at discharge and their discharge diagnosis was abstracted from their medical records by the study nurses. Following discharge, field officers contacted subjects over telephone at two- and four- months, and in-person at six-months to obtain vital status as well as any readmission events. All data was collected using encrypted study tablets and later uploaded to a Research Electronic Data Capture (REDCap) database hosted at the BC Children's Hospital Research Institute (Vancouver, BC, Canada) [21,22].

## Primary outcome and exposure variables

The goal of this secondary analysis was to use simple and multivariate Poisson regression to determine the unadjusted and adjusted risk of post-discharge mortality among participants based on single or dual exposure to pollutant fuel sources for cooking and household lighting. The primary outcome was defined as mortality up to 6-months post-discharge.

We created a proxy exposure variable to classify whether each participant was exposed to a pollutant fuel source for cooking, household lighting, or both using variables which recorded the type of cooking fuel used, location of cooking (indoors vs outdoors), and primary source of household lighting (see S1 Table). Participants whose guardians used charcoal, grass/shrubs, kerosene, paraffin, wood to cook were categorized as having been exposed to a pollutant cooking fuel source whereas the use of electricity, biogas, and propane was categorized as being exposed to a non-pollutant cooking fuel source. If the participant's household used kerosene lamps, tadoobas, and candles as their primary source of household lighting, then they were categorized as having been exposed to a pollutant source of household lighting. Conversely, the use of solar lanterns, solar powered bulbs, battery powered lights, and electric bulbs was categorized as a non-pollutant source of household lighting exposure. The proxy exposure variable was stratified into three exposure levels: dual exposure to both pollutant fuel sources for cooking indoors and household lighting; single exposure to either pollutant fuel sources used for household light or cooking indoors; and minimal exposure to pollutant fuel sources for cooking and household lighting.

## Statistical analysis

A Chi-Squared and Kruskal-Wallis test was used to assess the distribution of the potential confounders by exposure. The analysis consisted of using Poisson models with robust standard errors. We checked for outliers, multicollinearity between variables, and plotted residuals to check for independence. The multivariate Poisson regression model adjusted for age, sex, distance of home from facility, maternal education, and maternal HIV status. The analysis further assessed the effect of possible interactions between the exposure variable and variables such as age, sex, maternal education, and discharge diagnosis on the risk of post-discharge mortality. To test for interaction, models with and without the interaction term were compared using the likelihood ratio test. A p-value of < 0.05 was used to determine whether interaction terms were statistically significant.

Mortality 6-months post-discharge was treated as a binary outcome. Those who died in-hospital or did not complete follow-up were excluded from the analysis. Results were summarized using risk ratios with their corresponding 95% confidence intervals and p-values. A p-value less than 0.05 was used to determine statistical significance. Missing values were low and imputed using the k-nearest neighbour's approach. Descriptive data, such as medians with IQRs for continuous variables and counts with percentages for categorical variables were used to describe the population. The analysis was conducted in RStudio version 2021.09.0 (RStudio, Boston, MA, USA).

## Inclusivity in global research

Additional information regarding the ethical, cultural, and scientific considerations specific to inclusivity in global research is included in the Supporting Information (S5 Checklist).

## Results

7505 participants were enrolled: 3667 in the 0–6-month cohort and 3838 in the 6–60-month cohort. 7100 participants were discharged from the hospital alive, of which 145 were lost to follow-up at 6 months. A total of 6955 participants were included in the analysis. Table 1 provides the baseline characteristics of the study population.

Among the cohort aged 0–6 months, 263 of 3348 (7.9%) died after hospital discharge. Among the cohort aged 6–60 months, 176 of 3607 (4.9%) died after hospital discharge. This corresponds to an overall 6-month post-discharge mortality rate of 6.3%.

**Table 1. Baseline characteristics of the study population as a whole and stratified by single or dual exposure to pollutant fuel sources of cooking and lighting. Note: * beside the variable name indicates that the distribution of the potential confounder by exposure is statistically significant, as calculated by a Chi-Squared or a Kruskal-Wallis test. A p-value less than 0.05 was used to determine statistical significance.**

| Variable | Pollutant Cooking and Lighting Fuel Exposure Category | | | |
|---|---|---|---|---|
| | Total (n=6955) | Dual (n=1107 (15.9%)) | Single (n=4382 (63.0%)) | Minimal (n=1466 (21.1%)) |
| Mortality (%) | | | | |
| Alive | 6516 (93.7%) | 1017 (91.9%) | 4109 (93.8%) | 1390 (94.8%) |
| Deceased | 439 (6.3%) | 90 (8.1%) | 273 (6.2%) | 76 (5.2%) |
| Sex (%) | | | | |
| Female | 3073 (44.2%) | 500 (45.2%) | 1920 (43.8%) | 653 (44.5%) |
| Male | 3882 (55.8%) | 607 (54.8%) | 2462 (56.2%) | 813 (55.5%) |
| *Age, months (median (IQR)) | 6.8 (1.7 to 17.5) | 9.8 (2.5 to 22.2) | 6.1 (1.6 to 17.0) | 6.1 (1.4 to 15.1) |
| *Hospital of admission (%) | | | | |
| Mbarara Regional Referral Hospital | 1136 (16.3%) | 247 (22.3%) | 663 (15.1%) | 226 (15.4%) |
| Holy Innocents Children's Hospital | 1409 (20.3%) | 137 (12.4%) | 1030 (23.5%) | 242 (16.5%) |
| Masaka Regional Referral Hospital | 1455 (20.9%) | 222 (20.1%) | 861 (19.6%) | 372 (25.4%) |
| Jinja Regional Referral Hospital | 2322 (33.4%) | 392 (35.4%) | 1367 (31.2%) | 563 (38.4%) |
| Villa Maria Hospital | 201 (2.9%) | 29 (2.6%) | 142 (3.2%) | 30 (2.0%) |
| Uganda Martyrs Hospital | 432 (6.2%) | 80 (7.2%) | 319 (7.3%) | 33 (2.3%) |
| *Distance of home to facility, km (median (IQR) | 14.5 (4.6 to 31.5) | 19.0 (8.1 to 32.0) | 16.5 (5.1 to 33.3) | 5.5 (2.9 to 23.5) |
| *Maternal age, years (median (IQR) | 26 (23 to 30) | 26 (22.0 to 30.5) | 26 (23.0 to 30.5) | 25 (22.0 to 29.0) |
| *Maternal education (%) | | | | |
| No school | 264 (3.8%) | 63 (5.7%) | 170 (3.9%) | 31 (2.1%) |
| ≤P3 | 435 (6.3%) | 131 (11.8%) | 245 (5.6%) | 59 (4.0%) |
| P4 to P7 | 2873 (41.3%) | 592 (53.5%) | 1783 (40.7%) | 498 (34.0%) |
| S1 to S6 | 2516 (36.2%) | 277 (25.0%) | 1586 (36.2%) | 653 (44.5%) |
| Post secondary | 809 (11.6%) | 31 (2.8%) | 558 (12.7%) | 220 (15.0%) |
| Don't know | 58 (0.8%) | 13 (1.2%) | 40 (0.9%) | 5 (0.3%) |
| *Maternal HIV status (%) | | | | |
| Negative | 6076 (87.4%) | 907 (81.9%) | 3852 (87.9%) | 1317 (89.8%) |
| Positive | 549 (7.9%) | 107 (9.7%) | 329 (7.5%) | 113 (7.7%) |
| Unknown | 330 (4.7%) | 93 (8.4%) | 201 (4.6%) | 36 (2.5%) |
| Admission anthropometry | | | | |
| MUAC, mm* | 129 (111 to 142) | 130 (111 to 142) | 129 (111 to 142) | 129 (111 to 142) |
| <110 or <115 | 1545 (22.2%) | 255 (23.0%) | 953 (21.7%) | 337 (23.0%) |
| 110–120 or 115–125 | 1339 (19.3%) | 207 (18.7%) | 861 (19.6%) | 271 (18.5%) |
| >120 or >125 | 4071 (58.5%) | 645 (58.3%) | 2568 (58.6%) | 858 (58.5%) |
| *Weight-for-age Z score | −0.9 (−2.1 to 0.0) | −1.3 (−2.5 to −0.3) | −0.9 (−2.0 to 0.1) | −0.8 (−1.9 to 0.1) |
| <−3 | 954 (13.7%) | 198 (17.9%) | 590 (13.5%) | 166 (11.3%) |
| −3 to −2 | 923 (13.3%) | 196 (17.7%) | 548 (12.5%) | 179 (12.2%) |
| >−2 | 5078 (73.0%) | 713 (64.4%) | 3244 (74.0%) | 1121 (76.5%) |
| *Length-for-age Z score | −0.7 (−2.0 to 0.5) | −1.0 (−2.3 to 0.2) | −0.7 (−1.9 to 0.5) | −0.6 (−1.8 to 0.6) |
| <−3 | 838 (12.0%) | 161 (14.5%) | 521 (11.9%) | 156 (10.6%) |
| −3 to −2 | 861 (12.4%) | 164 (14.8%) | 522 (11.9%) | 175 (11.9%) |
| >−2 | 5256 (75.6%) | 782 (70.6%) | 3339 (76.2%) | 1135 (77.4%) |
| *BMI Z score | −0.9 (−2.2 to 0.3) | −1.2 (−2.6 to 0.1) | −0.9 (−2.2 to 0.3) | −0.7 (−2.1 to 0.2) |
| <−3 | 1113 (16.0%) | 219 (19.8%) | 691 (15.8%) | 203 (13.8%) |
| −3 to −2 | 887 (12.8%) | 170 (15.4%) | 538 (12.3%) | 179 (12.2%) |

*(Continued)*

**Table 1.** (Continued)

| Variable | Pollutant Cooking and Lighting Fuel Exposure Category | | | |
| | Total (n = 6955) | Dual (n = 1107 (15.9%)) | Single (n = 4382 (63.0%)) | Minimal (n = 1466 (21.1%)) |
|---|---|---|---|---|
| >–2 | 4955 (71.2%) | 718 (64.9%) | 3153 (72.0%) | 1084 (73.9%) |
| *Weight-for-length Z score | −0.9 (−2.3 to 0.3) | −1.1 (−2.6 to 0.1) | −0.8 (−2.2 to 0.4) | −0.8 (−2.1 to 0.2) |
| <–3 | 1145 (16.5%) | 233 (21.0%) | 694 (15.8%) | 218 (14.9%) |
| –3 to –2 | 879 (12.6%) | 150 (13.6%) | 544 (12.4%) | 185 (12.6%) |
| >–2 | 4931 (70.9%) | 724 (65.4%) | 3144 (71.7%) | 1063 (72.5%) |
| Discharge diagnosis | | | | |
| *Malaria | 1381 (19.9%) | 313 (28.3%) | 840 (19.2%) | 228 (15.6%) |
| Pneumonia | 2232 (32.1%) | 373 (33.7%) | 1387 (31.7%) | 472 (32.2%) |
| *Bronchiolitis | 354 (5.1%) | 60 (5.4%) | 240 (5.5%) | 54 (3.7%) |
| *Upper respiratory tract infection | 520 (7.5%) | 63 (5.7%) | 355 (8.1%) | 102 (7.0%) |
| Reactive airway disease or asthma | 37 (0.5%) | 3 (0.3%) | 26 (0.6%) | 8 (0.5%) |
| *Gastroenteritis or diarrhea | 1009 (14.5%) | 147 (13.3%) | 606 (13.8%) | 256 (17.5%) |
| *HIV-related or AIDS-related disease | 61 (0.9%) | 24 (2.2%) | 24 (0.5%) | 13 (0.9%) |
| Meningitis or encephalitis | 242 (3.5%) | 32 (2.9%) | 168 (3.8%) | 42 (2.9%) |
| *Malnutrition | 452 (6.5%) | 121 (10.9%) | 252 (5.8%) | 79 (5.4%) |
| *Tuberculosis | 86 (1.2%) | 22 (2.0%) | 43 (1.0%) | 21 (1.4%) |
| *Skin or soft-tissue infection | 227 (3.3%) | 28 (2.5%) | 164 (3.7%) | 35 (2.4%) |
| Measles | 419 (6.0%) | 72 (6.5%) | 252 (5.8%) | 95 (6.5%) |
| *Sepsis | 2209 (31.8%) | 286 (25.8%) | 1393 (31.8%) | 530 (36.2%) |
| Genetic or congenital disease | 152 (2.2%) | 22 (2.0%) | 97 (2.2%) | 33 (2.3%) |
| Sickle cell anemia | 64 (0.9%) | 15 (1.4%) | 38 (0.9%) | 11 (0.8%) |
| *Febrile seizure | 24 (0.3%) | 2 (0.2%) | 12 (0.3%) | 10 (0.7%) |
| Other infection | 143 (2.1%) | 23 (2.1%) | 89 (2.0%) | 31 (2.1%) |
| Other non-infection | 295 (4.2%) | 44 (4.0%) | 191 (4.4%) | 60 (4.1%) |
| *Positive malaria test | 1391 (20%) | 312 (28.2%) | 849 (19.4%) | 230 (15.7%) |
| *HIV status | | | | |
| Positive | 206 (3.0%) | 44 (4.0%) | 114 (2.6%) | 48 (3.3%) |
| Negative | 6749 (97.0%) | 1063 (96.0%) | 4268 (97.4%) | 1418 (96.7%) |
| *Haemoglobin status | 12 (10 to 14) | 11.3 (9.3 to 13.3) | 12.0 (10.3 to 14.0) | 12.7 (10.7 to 14.0) |
| Not anaemic: ≥ 11 g/dL | 4593 (66.0%) | 630 (56.9%) | 2916 (66.5%) | 1047 (71.4%) |
| Mild anaemia: 7–10 g/dL | 1814 (26.1%) | 330 (29.8%) | 1136 (25.9%) | 348 (23.7%) |
| Severe anaemia: < 7 g/dL | 548 (7.9%) | 147 (13.3%) | 330 (7.5%) | 71 (4.8%) |

Table 2 describes the distribution of the variables used to create the proxy exposure variable. Parents or guardians of 6936 of the participants mentioned using at least one type of pollutant fuel source to cook. Of the 6955 participants, 5122 (73.6%) lived in households where cooking primarily occurred indoors and 1492 (21.5%) lived in households where a pollutant was used as the primary source of household lighting. After creating the proxy exposure variable, 1107 participants (15.9%) had dual exposure to both a pollutant fuel source for cooking and household lighting, 4382 (63.0%) had a single exposure to either a pollutant fuel source for cooking or household lighting, and 1466 (21.1%) participants did not have a single exposure to a pollutant fuel source for cooking and household lighting.

The overall 6-month post-discharge mortality was 8.1%, 6.2%, and 5.2% for individuals in the dual exposure, single exposure, and minimal exposure category, respectively. The results from the simple and multivariate Poisson regression models exploring the risk of 6-month post-discharge mortality by dual or single exposure to pollutant fuel sources

**Table 2. Distribution of variables used to create the proxy exposure variable.**

| Variable | Total (%) |
|---|---|
| Type of cooking fuel used | |
| Pollutant | 6936 (99.7) |
| Non-pollutant | 19 (0.3) |
| Location of cooking | |
| Indoors | 5122 (73.6) |
| Outdoors | 1833 (26.4) |
| Primary source of household lighting | |
| Pollutant | 1492 (21.5) |
| Non-pollutant | 5463 (78.5) |

**Table 3. Unadjusted and adjusted risk ratios comparing the risk of death within 6-months post-discharge for individuals with dual or single exposure to pollutant fuel sources for cooking and household lighting compared to minimal.**

| Dual or single exposure | Unadjusted Risk Ratio (95% CI) | Adjusted Risk Ratio[1] (95% CI) |
|---|---|---|
| Dual[2] | 1.57 (1.17, 2.11) | 1.30 (0.96, 1.76) |
| Single[2] | 1.20 (0.94, 1.54) | 1.08 (0.84, 1.38) |

*Abbreviations*: CI, confidence interval.

[1] The multivariable model adjusts for age, sex, distance of home to facility, maternal education, and maternal HIV status.

[2] Reference group: Individuals categorized as having minimal exposure to pollutant fuel sources for cooking and household lighting.

for cooking indoors and household lighting is summarized in Table 3. In the unadjusted model, the risk of 6-month post-discharge mortality among with dual exposure was 1.57 (95% CI: 1.17, 2.11) times the risk for those with minimal exposure to pollutant fuel sources for cooking indoors and household lighting. Furthermore, the unadjusted risk of 6-month post-discharge mortality for those with single exposure was 1.20 (95% CI: 0.94, 1.54) times the risk for those with minimal exposure. In the adjusted model, the risk of 6-month post-discharge mortality decreased for both dual (adjusted RR = 1.30, 95% CI: 0.96, 1.76) and single (adjusted RR: 1.08, 95% CI: 0.84, 1.38) exposure when compared to minimal. The adjusted model controlled for age, sex, distance of home to facility, maternal education, and maternal HIV status. After checking for effect modification, there was no significant interaction between exposure and age, sex, or maternal education. We also conducted several sensitivity analyses which consisted of stratifying the sample population according to their discharge diagnoses. We then explored the adjusted risk of 6-month post-discharge mortality by the exposure variable among each stratum. These sensitivity analyses did not yield any statistically significant findings. The output of these analyses is outlined in S2 Table. We also stratified our analysis by anthropometry, restricting the analysis to those with weight-for-age z-scores < −2, and found similar results.

## Discussion

In this prospective, multi-site observational study we found high rates of exposure to pollutant fuel sources within the home environment among children who were hospitalized with suspected sepsis. Despite the high mortality rate during the post-discharge period, we did not observe that children residing in home environments with either single or dual exposure to pollutant fuel sources experienced a statistically significant higher risk of death. However, these results do not suggest that exposure to pollutants have no effect on child health. In rural Uganda there are many factors that contribute

to poor health outcomes such as poverty, food insecurity leading to malnutrition, and comorbidities such as anemia, HIV infection and malaria. The effects of sociodemographic disadvantages including indoor pollutants are likely additive and therefore it may be difficult to tease apart the individual contributions.

The relationship between household air pollution and mortality 6-month post discharge may still be confounded by other factors such as anthropometry, social vulnerability, and illness acuity. For example, lower socioeconomic class, lower anthropometric measures, and diagnoses such as malnutrition and anemia are reported to be associated with post-discharge mortality [16,23]. In our analysis, these three factors were found to be associated with both exposure and with death in the short term. Furthermore, the impact of air pollution is typically measured over prolonged periods of time since outcomes are typically rare and require prolonged periods of exposure to manifest themselves [24]. Within the context of this study, where mortality occurred at frequencies far exceeding community levels, the relative impact of comorbidities, social vulnerabilities and features of the prior illness may play a more acute role on mortality than environmental factors. Thus, a relevant question of whether exposure to particulate matter from cooking and household lighting practices contribute to mortality beyond the 6-month study period, when community-level mortality returns to typical levels, cannot be answered by this study.

Among the nearly 7000 admitted children, over 99% of caregivers reported using at least one type of pollutant fuel source to cook, highlighting the reliance families have on these sources of energy. Cooking with pollutant fuel is more likely among the poor and hence should be included among the factors associated with sociodemographic vulnerability. Other factors such as traffic and industrial emission also contribute to ambient air pollution levels which subsequently invade indoor living spaces may contribute more to adverse health because of chronic exposure [25]. Therefore, addressing ambient air pollution is also important but has not been taken into account when exploring the effect of household air pollution on mortality.

Current studies exploring household air pollution commonly use proxy variables such as type of cooking fuel and cooking location when exploring impacts on all-cause mortality [8–14,26–29]. Other studies have used a more direct measurement of exposure such as directly determining the concentration of PM2.5 in indoor living spaces [27,30,31]. But there is incongruence between the results of these studies. For example, three studies reported an increase in all-cause mortality due to indoor cooking whereas one found no statistically significant association between this exposure and outcome [9,12,13,29]. The same goes for type of cooking fuel used and indoor PM2.5 levels, whereby some studies found a statistically significant association with mortality while others did not [8–14,26–28,30,31]. As a result, determining the casual relationship between household air pollution and short-term mortality is not straightforward, even with access to PM2.5 readings. In addition, several of the studies did not control for confounding, many were cross sectional in nature, most did not measure direct PM2.5 exposure, and most did not address ambient air pollution exposure.

This study is subject to several limitations. First, this analysis used self-reported/indirect measures of exposure, specifically, cooking and household lighting. The degree of pollutant exposure from cooking and lighting may differ from household to household due to physical characteristics of the home such as degree of ventilation and location of cookstove in the home, and due to differences in cooking practices at the familial level. Furthermore, self-reported measures captured at a single time point are not as reliable and accurate as more direct measures especially for particulate matter exposure in one's home. To truly determine the impact of these cooking and lighting practices on health outcomes, direct measures of particulate matter in individual households should be used to determine its association with 6-month post-discharge mortality. Future studies should incorporate more objective measures of HAP, such as devices which directly measure PM2.5 exposure, to more accurately measure the relationship between HAP and post-discharge outcomes. Second, although this analysis controlled for age, sex, and proxies of socioeconomic status, the reported non-significant trend towards increasing risk of post-discharge mortality that we observe when going from minimal, single, to dual exposure may still be confounded by factors such as ambient air pollution, social vulnerability, and illness acuity. Other characteristics such as building ventilation, cooking frequency, and location of child may also confound the true relationship between

household air pollution and post-discharge mortality. However, recognizing that as many children die following discharge as during the acute phase of illness, the impact of HAP on this phase of recovery has not previously been explored, and regardless of the typical limitations to HAP exposure measurement that may exist in our analysis, these results must spur further investigations into the increasingly topical and relevant area of the post-acute illness period. It is this distinction which makes our study unique and relevant.

## Conclusion

In conclusion, this analysis did not find a statistically significant association between household air pollution and 6-month post-discharge mortality among children under the age of 5 years who were admitted to hospital in Uganda with a severe infection. However, this does not suggest that household air pollution has no effect on post-discharge mortality within this population due to the exposure being compounded by other major drivers of mortality such as social vulnerabilities, illness acuity, and ambient air pollution exposure. Future research should explore this association using more objective measurements of household air pollution over a longer period of time.

## Supporting information

**S1 Table. Distribution of proxy exposure variable by type of cooking fuel, cooking location, primary source of household lighting, and mortality.**
(DOCX)

**S2 Table. Stratified analysis categorized by discharge diagnosis exploring adjusted risk ratios comparing the risk of death within 6-months post-discharge for individuals with dual or single exposure to pollutant fuel sources for cooking and household lighting compared to minimal.**
(DOCX)

**S3 Table. Full results of the bivariate Poisson regression analyses performed.**
(DOCX)

**S4 Table. Full results of the multivariate Poisson regression analyses performed.**
(DOCX)

**S5 Checklist. Inclusivity in global research questionnaire.**
(DOCX)

## Acknowledgments

We gratefully acknowledge patient participants and staff of the Smart Discharges in Uganda, without whom this study would not have been possible.

## Author contributions

**Conceptualization:** Gurvir S. Dhutt, Cherri Zhang, Peter Moschovis, Niranjan Kissoon, Matthew O. Wiens.

**Data curation:** Gurvir S. Dhutt, Cherri Zhang.

**Formal analysis:** Gurvir S. Dhutt, Cherri Zhang.

**Investigation:** Gurvir S. Dhutt, Elias Kumbakumba, Abner Tagoola, Peter Moschovis, Stephen Businge, Nathan Kenya Mugisha, Jerome Kabakyenga.

**Methodology:** Gurvir S. Dhutt, Cherri Zhang, Niranjan Kissoon, Matthew O. Wiens.

**Project administration:** Gurvir S. Dhutt, Elias Kumbakumba, Abner Tagoola, Stephen Businge, Nathan Kenya Mugisha, Jerome Kabakyenga, Matthew O. Wiens.

**Supervision:** Cherri Zhang, Elias Kumbakumba, Abner Tagoola, Stephen Businge, Niranjan Kissoon, Matthew O. Wiens.

**Validation:** Gurvir S. Dhutt, Cherri Zhang.

**Writing – original draft:** Gurvir S. Dhutt.

**Writing – review & editing:** Gurvir S. Dhutt, Cherri Zhang, Elias Kumbakumba, Abner Tagoola, Peter Moschovis, Stephen Businge, Niranjan Kissoon, Nathan Kenya Mugisha, Jerome Kabakyenga, Matthew O. Wiens.

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
