## [Editor Report · Decision Letter 0]

PONE-D-24-52954Exposure to pollutants for household cooking and lighting and pediatric post-discharge mortality following a severe infection in UgandaPLOS ONE

Dear Dr. Dhutt,

Thank you for submitting your manuscript to PLOS ONE. After careful consideration, we feel that it has merit but does not fully meet PLOS ONE’s publication criteria as it currently stands. Therefore, we invite you to submit a revised version of the manuscript that addresses the points raised during the review process.

Editor's Comments:Major revision required. Please see the comments below and revise the paper accordingly.==============================

We look forward to receiving your revised manuscript.

Kind regards,

Srijan Lal Shrestha, Ph.D.

Academic Editor

PLOS ONE

Journal Requirements:

2. Please include a complete copy of PLOS’ questionnaire on inclusivity in global research in your revised manuscript. Our policy for research in this area aims to improve transparency in the reporting of research performed outside of researchers’ own country or community. The policy applies to researchers who have travelled to a different country to conduct research, research with Indigenous populations or their lands, and research on cultural artefacts. The questionnaire can also be requested at the journal’s discretion for any other submissions, even if these conditions are not met.  

Please find more information on the policy and a link to download a blank copy of the questionnaire here: https://journals.plos.org/plosone/s/best-practices-in-research-reporting. Please upload a completed version of your questionnaire as Supporting Information when you resubmit your manuscript.

Additional Editor Comments :

I went through the submitted paper entitled: “Exposure to pollutants for household cooking and lighting and pediatric post-discharge mortality following a severe infection in Uganda”

The paper tries to establish relationship between air pollution exposure from household cooking and lighting and pediatric post-discharge mortality following severe infection based upon prospective cohort design. However, the results of the study shows absence of such relationship accounting other variables. I have some major and minor comments before the paper can be forwarded for peer review.

Major comments

1. Even though the research was conducted to establish relationship between HAP exposure and child mortality, it showed otherwise. In this context, it is very essential to discuss about the importance about the study. Authors have mentioned lots of the limitations of the study. It also requires to justify the benefits of the study. Consequently, why is it important to publish the findings of the study in the presence of so many limitations in the study? This needs to be clarified and discussed.

2. Introduction part is very short for a research paper which has such an important issue. Authors should include important and related literature reviews which have quantification of effects of HAP on childhood mortality.

3. Why this research based upon only 6 hospital patients is representative of Uganda? How many critical child caring hospitals are there in Uganda? This is a critical question which needs to be justified. Otherwise, mentioning of Uganda will not be accepted. Instead, authors should mention the representation of results for only the selected sub-groups or regions, if this is the case.

4. Authors have used logistic regression for analysis by specifying only the adjusted odds ratios. Instead, it is suggested to specify the full logistic regression model with all the explanatory variables included along with their estimates and relevant statistics. Also, how were the interaction effects assessed? It needs to be clarified.

5. Assumptions and model adequacy tests required for the logistic model have not been mentioned in the paper. This is also essential in modeling perspective.

Minor comments

1. It is better to mention about inclusion and exclusion criteria of the research. If mentioned, it should be better in this sub-heading.

2. Why 2 citations of data collection method? Only the citation whose data collection method matches exactly with the research needs to be mentioned.

3. Elaborate on statement that the analysis did not adjust for multiple comparisons.

4. Role of funding source/ ethical clearance need not be mentioned in the main text of a paper. They will be mentioned as per the journal guidelines.

5. Table 1: It is also required to mention about the mortality status along with exposure and other variables.

6. Why relative risks (RRs) are not reported for a prospective cohort study results?

Authors are requested to address all the comments raised. Failing to address the comments appropriately in the revision may lead to rejection of the paper.
---

## [Author Response · Author response to Decision Letter 1]

20 Feb 2025

Hello Dr. Srijan Lal Shrestha,

Thank you for your thorough review of our manuscript and the constructive comments that you provided. We appreciate the time and effort you have invested in assessing our work.

Our team has addressed all comments and additional journal requirements detailed in the decision email. We have provided our responses to all major/minor comments below in an orderly fashion. We have also ensured that our manuscript meets PLOS ONE’s style requirements and uploaded a complete copy of PLOS’ questionnaire on inclusivity in global research. Regarding data sharing, due to the sensitive nature of the patient data and the risk for re-identification, data is only available through reasonable request to the Pediatric Sepsis Data CoLaboratory. All data and collection tools are available through the Smart Discharges Dataverse. The dataset is published and can be viewed using this link: https://doi.org/10.5683/SP3/ZLZQOG (Citation: Dhutt GS, Zhang C, Kumbakumba E, Tagoola A, Moschovis P, Businge S, et al. Exposure to pollutants for household cooking and lighting and pediatric post-discharge mortality following a severe infection in Uganda [Internet]. V1 ed. Borealis; 2025. Available from: https://doi.org/10.5683/SP3/ZLZQOG ). Access to the data can be requested through the link above. Access to data is granted on a case-by-case basis following approval from the authors and the Data Governance Committees.

Note: page and line numbers are provided for new text which have been added to the manuscript. The page and line numbers correspond to those in the manuscript document with tracked changes titled “Revised Manuscript with Track Changes”.

We hope our revisions meet your expectations, and we are happy to address any further concerns. Thank you for considering our manuscript for publication.

Major comments

1. Even though the research was conducted to establish relationship between HAP exposure and child mortality, it showed otherwise. In this context, it is very essential to discuss about the importance about the study. Authors have mentioned lots of the limitations of the study. It also requires to justify the benefits of the study. Consequently, why is it important to publish the findings of the study in the presence of so many limitations in the study? This needs to be clarified and discussed.

Thank you for this comment. We agree that our work showed otherwise however, our work uses a similar methodology for exposure data as many other research studies that show both similar and divergent results (see lines 287-299). This underscores our argument that using proxy variables may not be reliable and accurate. Regardless, our study is of general relevance to the research community in that it examines an area not previously explored, namely the impact of HAP on short term outcomes in the context of illness recovery and convalescence. Our research team focuses primarily on post-discharge recovery, recognizing that as many children die following discharge as during the acute phase of illness. The impact of HAP on this phase of recovery has not previously been explored, and regardless of the typical limitations to HAP exposure measurement that may exist in our analysis, these results must spur further investigation into the increasingly topical and relevant area of the post-acute illness period. It is this distinction makes our study unique and relevant.

We have included the following in our discussion:

Page #21, line 316-321 “However, recognizing that as many children die following discharge as during the acute phase of illness, the impact of HAP on this phase of recovery has not previously been explored, and regardless of the typical limitations to HAP exposure measurement that may exist in our analysis, these results must spur further investigations into the increasingly topical and relevant area of the post-acute illness period. It is this distinction which makes our study unique and relevant.”

2. Introduction part is very short for a research paper which has such an important issue. Authors should include important and related literature reviews which have quantification of effects of HAP on childhood mortality.

Thank you for your suggestion. We have included the following in the Introduction:

Page #4, line 80-83 “This practice results in harmful indoor air pollution and, in 2020, was reported to be associated with about 3.2 million deaths, disproportionately impacting children in Sub-Saharan Africa due to increased pollution levels resulting from rapid urbanization and industrialization in this region. (1-4)”

And on Page #5, line 88-93 “Children further face increased risk due to extended exposure to particulate matter from household air pollution from engagement in household activities such as cooking and household lighting. (1, 6) In the Sub-Saharan African context, the use of household pollutant fuel sources (e.g., kerosene or coal) has been shown to increase the risk of all-cause child mortality, with current estimates being as high as a 126% increase in mortality due to pollutant cooking fuel source use inside the household. (8-14)”

3. Why this research based upon only 6 hospital patients is representative of Uganda? How many critical child caring hospitals are there in Uganda? This is a critical question which needs to be justified. Otherwise, mentioning of Uganda will not be accepted. Instead, authors should mention the representation of results for only the selected sub-groups or regions, if this is the case.

We agree that not all hospitals in Uganda were used in the study. However, we feel that the hospitals are representative of Uganda in that the public and faith-based facilities included in this study consist of a catchment area comprising 30 districts with an approximate population of 8.2 million individuals, of which 1.4 million were children younger than 5 years at the time data collection was conducted. (See Uganda Bureau of Statistics. Population and censuses. 2022. https://www.ubos.org/explore-statistics/20/). Hence, we believe the 6 hospitals chosen are representative of Uganda as these hospitals represent the major facilities caring for critically ill children within the regions covered (Central Uganda, Southwest Uganda, Eastern Uganda). Having said this, we are happy to remove the mention of Uganda if the editor feels it is not sufficiently justified.

4. Authors have used logistic regression for analysis by specifying only the adjusted odds ratios. Instead, it is suggested to specify the full logistic regression model with all the explanatory variables included along with their estimates and relevant statistics. Also, how were the interaction effects assessed? It needs to be clarified.

Based on the feedback in the minor revisions section, we have reanalyzed our data using Poisson regression with robust standard errors and provided risk ratios as opposed to odds rations. The full Poisson regression model with all the explanatory variables and their estimates/relevant statistics are provided in the supplementary file titled “S3_Table; S4_Table”. The results section of the manuscript provides the unadjusted and adjusted risk ratios in Table 3, with a mention of all the variables that were included in the adjusted model.

We have added the following to the methods section regarding how the interaction effects were assessed:

Page 9, lines 180-182: “To test for interaction, models with and without the interaction term were compared using the likelihood ratio test. A p-value of < 0.05 was used to determine whether interaction terms were statistically significant.”

5. Assumptions and model adequacy tests required for the logistic model have not been mentioned in the paper. This is also essential in modeling perspective.

We have reanalyzed our data using Poisson regression with robust standard errors and provided risk ratios instead of odds ratios.

We have added the following to the methods section regarding how assumptions of the model were checked:

Page 9, lines 174-176: “The analysis consisted of using Poisson models with robust standard errors. We checked for outliers, multicollinearity between variables, and plotted residuals to check for independence.”

Minor comments

1. It is better to mention about inclusion and exclusion criteria of the research. If mentioned, it should be better in this sub-heading.

We have changed the “Study population” heading to “Study population inclusion and exclusion criteria” to reflect this.

2. Why 2 citations of data collection method? Only the citation whose data collection method matches exactly with the research needs to be mentioned.

We have removed the second citation and kept the citation whose data collection method matches exactly with the research.

3. Elaborate on statement that the analysis did not adjust for multiple comparisons.

This statement has been removed from the manuscript.

4. Role of funding source/ ethical clearance need not be mentioned in the main text of a paper. They will be mentioned as per the journal guidelines.

We have removed this section from the main text of the paper.

5. Table 1: It is also required to mention about the mortality status along with exposure and other variables.

We have added mortality status to Table 1.

6. Why relative risks (RRs) are not reported for a prospective cohort study results?

We have replaced the odds ratios with risk ratios in the manuscript.

---

## [Decision Letter · Decision Letter 1]

PONE-D-24-52954R1Exposure to pollutants for household cooking and lighting and pediatric post-discharge mortality following a severe infection in UgandaPLOS ONE

Dear Dr.  Dhutt,

Thank you for submitting your manuscript to PLOS ONE. After careful consideration, we feel that it has merit but does not fully meet PLOS ONE’s publication criteria as it currently stands. Therefore, we invite you to submit a revised version of the manuscript that addresses the points raised during the review process.

We look forward to receiving your revised manuscript.

Kind regards,

Srijan Lal Shrestha, Ph.D.

Academic Editor

PLOS ONE

Journal Requirements:

Additional Editor Comments:

Minor comments have been received from reviewers. Authors need to revise the manuscript accordingly.

Specific reviewer comment (along with the attached comments) is as follows.

The future studies should use objective measures like indoor air sensors for PM2.5 to strengthen causality claims. Only 6 hospitals were included, even though they serve a large area. So, acknowledging the regional diversity within Uganda; broader national sampling could be used for claiming representativeness. Although interaction terms were tested, deeper analysis (e.g., stratified results by nutritional status) could reveal subgroup vulnerabilities' 6-month follow-up may not capture chronic effects of HAP. The longer-term studies are needed to observe delayed health impacts. While results are inconclusive, the study could better link findings to policy implications, like clean fuel promotion or post-discharge care programs. More discussion could be included on how cooking practices are culturally rooted, and how interventions should be community-sensitive.

Reviewers' comments:

Reviewer's Responses to Questions

**Comments to the Author**

1. If the authors have adequately addressed your comments raised in a previous round of review and you feel that this manuscript is now acceptable for publication, you may indicate that here to bypass the “Comments to the Author” section, enter your conflict of interest statement in the “Confidential to Editor” section, and submit your "Accept" recommendation.

Reviewer #1: (No Response)

Reviewer #2: All comments have been addressed

2. Is the manuscript technically sound, and do the data support the conclusions?

Reviewer #1: Yes

Reviewer #2: Partly

3. Has the statistical analysis been performed appropriately and rigorously? 

Reviewer #1: I Don't Know

Reviewer #2: No

4. Have the authors made all data underlying the findings in their manuscript fully available?

Reviewer #1: Yes

Reviewer #2: Yes

5. Is the manuscript presented in an intelligible fashion and written in standard English?

Reviewer #1: Yes

Reviewer #2: Yes

6. Review Comments to the Author

Reviewer #1: Authors have addressed the comments provided by the previous reviewer. Since the result did not showed relationship with HAP exposure and child mortality which is against established evidence from many studied. Therefore, further discussion on contextual factors and reasons is desirable.

Reviewer #2: The future studies should use objective measures like indoor air sensors for PM2.5 to strengthen causality claims. Only 6 hospitals were included, even though they serve a large area. So, acknowledging the regional diversity within Uganda; broader national sampling could be used for claiming representativeness. Although interaction terms were tested, deeper analysis (e.g., stratified results by nutritional status) could reveal subgroup vulnerabilities' 6-month follow-up may not capture chronic effects of HAP. The longer-term studies are needed to observe delayed health impacts. While results are inconclusive, the study could better link findings to policy implications, like clean fuel promotion or post-discharge care programs. More discussion could be included on how cooking practices are culturally rooted, and how interventions should be community-sensitive.

7. PLOS authors have the option to publish the peer review history of their article (what does this mean? ). If published, this will include your full peer review and any attached files.

**Do you want your identity to be public for this peer review?** For information about this choice, including consent withdrawal, please see our Privacy Policy .

Reviewer #1: No

Reviewer #2: **Yes: ** Samikshya Acharya

---

## [Author Response · Author response to Decision Letter 2]

5 May 2025

Reviewer #1 Comment: Authors have addressed the comments provided by the previous reviewer. Since the result did not showed relationship with HAP exposure and child mortality which is against established evidence from many studied. Therefore, further discussion on contextual factors and reasons is desirable.

Thank you for this comment. We completely agree and recognize that it is important to situate our findings based on contextual factors in the hopes that it spurs future research to further explore our area of interest. While we agree that a significant body of evidence has linked HAP with childhood mortality, very little work has been done among children at high risk of mortality over the short term (i.e. 6 months post-discharge as in this study). We also agree that objective measures of PM2.5 exposure would have added a further element of robustness to our analysis and we have included this limitation in our discussion. We believe that this research justifies the exploration of long-term impacts of HAP on mortality following acute illness to determine if, over time, the effects of air pollution are additive.

Our limitations section already outlines exposure measurement and confounding as two key contextual factors relevant to our results. To add additional clarity, we have added another sentence to the limitations which states that:

Page #21, line 313-315 “Future studies should incorporate more objective measures of HAP, such as devices which directly measure PM2.5 exposure, to more accurately measure the relationship between HAP and post-discharge outcomes.

With regards to deviations from existing literature, we have emphasized the unique nature of this study which makes it novel. We state:

Page #21, line 321-326 “Recognizing that as many children die following discharge as during the acute phase of illness, the impact of HAP on this phase of recovery has not previously been explored, and regardless of the typical limitations to HAP exposure measurement that may exist in our analysis, these results must spur further investigations into the increasingly topical and relevant area of the post-acute illness period. It is this distinction which makes our study unique and relevant.”

We hope that this properly contextualizes our results.

Reviewer #2 Comment: The future studies should use objective measures like indoor air sensors for PM2.5 to strengthen causality claims. Only 6 hospitals were included, even though they serve a large area. So, acknowledging the regional diversity within Uganda; broader national sampling could be used for claiming representativeness. Although interaction terms were tested, deeper analysis (e.g., stratified results by nutritional status) could reveal subgroup vulnerabilities' 6-month follow-up may not capture chronic effects of HAP. The longer-term studies are needed to observe delayed health impacts. While results are inconclusive, the study could better link findings to policy implications, like clean fuel promotion or post-discharge care programs. More discussion could be included on how cooking practices are culturally rooted, and how interventions should be community-sensitive.

Thank you for this comment. It is extremely important to conduct a deeper analysis into how subgroup vulnerabilities may impact exposure and outcome. For this reason, we stratified our analysis by restricting the analysis to those with weight-for-age z-scores <-2 and found similar results. We have included the following in the results section:

Page #17, lines 241-242 “We also stratified our analysis by anthropometry, restricting the analysis to those with weight-for-age z-scores <-2, and found similar results.”

Furthermore, we agree that it is important to clearly outline how future studies can build on our work and put this into action items. To add additional clarity, we have added another sentence to the limitations which states that:

Page #21, line 313-315 “Future studies should incorporate more objective measures of HAP, such as devices which directly measure PM2.5 exposure, to more accurately measure the relationship between HAP and post-discharge outcomes.

With regards to promotion of clean fuel promotion in post-discharge care programs or other policy implications, we do not feel that these naturally flow from the results of this analysis. While we agree that such programs have ample justification from other research, linking them to this study seems somewhat problematic given our results. However, we do emphasize in our discussion the need for more work, especially related to longer term outcomes and more accurate measures of exposure measurement.

---

## [Editor Report · Decision Letter 2]

Exposure to pollutants for household cooking and lighting and pediatric post-discharge mortality following a severe infection in Uganda

PONE-D-24-52954R2

Dear Dr. Dhutt

We’re pleased to inform you that your manuscript has been judged scientifically suitable for publication and will be formally accepted for publication once it meets all outstanding technical requirements.

Kind regards,

Srijan Lal Shrestha, Ph.D.

Academic Editor

PLOS ONE
---

## [Editor Report · Acceptance letter]

PONE-D-24-52954R2

PLOS ONE

Dear Dr. Wiens,

I'm pleased to inform you that your manuscript has been deemed suitable for publication in PLOS ONE. Congratulations! Your manuscript is now being handed over to our production team.

Kind regards,

on behalf of

Dr. Srijan Lal Shrestha

Academic Editor

PLOS ONE